# Brain Tumor-Induced Changes in Routine Parameters of the Lipid Spectrum of Blood Plasma and Its Short-Chain Fatty Acids

**DOI:** 10.3390/cimb47040228

**Published:** 2025-03-26

**Authors:** Larisa Obukhova, Natalia Shchelchkova, Igor Medyanik, Konstantin Yashin, Artem Grishin, Oksana Bezvuglyak, Ilkhom Abdullaev

**Affiliations:** Federal State Budgetary Educational Institution of Higher Education, The “Privolzhsky Research Medical University” of the Ministry of Health of the Russian Federation, 603005 Nizhny Novgorod, Russia; n.shchelchkova@mail.ru (N.S.); med_neuro@inbox.ru (I.M.); yashinmed@yandex.ru (K.Y.); zhest8242@mail.ru (A.G.); bezvugliak.oksana@gmail.com (O.B.); ilhomabdullaev@yahoo.com (I.A.)

**Keywords:** blood plasma, lipids, cholesterol, neutral lipid/phospholipid ratio, short-chain fatty acids, butyric acid, molecular genetic markers of gliomas, Ki-67

## Abstract

The aim of this research was to provide a comparative analysis of the major parameters of the blood lipid spectrum found both in the case of brain tumors and in atherosclerosis, as well as to assess the correlation of these indicators with the proliferative activity index Ki-67 in cells. Blood analyses were conducted on samples from 50 patients with brain tumors and 50 patients with cerebral atherosclerosis. Blood plasma from 50 essentially healthy people was used for controls. Significant differences were found in the parameter values between the atherosclerosis sufferers and the control group only for their ratios of neutral lipids to cholesterol. Of the short-chain fatty acids, butyric acid is of greatest interest due to the significant differences of its levels from the control group in the blood of both patients with meningiomas and of those with gliomas. Statistically significant correlation coefficients between the levels of the Ki-67 cell proliferation marker and, in particular, butyric acid were found when compared with the neutral lipids to cholesterol ratios. These identified parameters of the blood plasma lipid spectrum can be used for preoperative diagnostics of brain tumors. However, these ratios cannot be used as preoperative noninvasive predictors of the level of the Ki-67 mitotic index, as no significant differences corresponding to this were found for low-grade or for high-grade anaplasia of brain tumors.

## 1. Introduction

The investigation of aberrations in the metabolome, in particular, of levels of those molecules involved in lipid metabolism is a promising research area in tumor diagnostics [1]. Compared to non-malignant tissues, glial tumors, for example, are characterized by overall increases in lipid levels and by high levels of lipids within these specific tumors [2]. However, lipid metabolism disorders are also typical of other diseases, primarily atherosclerosis [3].

Therefore, a new approach is needed to understand lipid metabolism in the pathogenesis of brain tumors, including the investigation of the levels of short-chain fatty acids (SCFAs). These are monocarboxylic acids with a chain length of up to six carbon atoms and are the main products of the anaerobic fermentation of indigestible polysaccharides, such as dietary fiber and resistant starch, being carried out by the microbiota in the colon [4]. As well as having an impact on the immune system, such SCFAs are involved in the regulation of neutrophil chemotaxis, the induction of regulatory T-cell development, IL-10 secretion, NF-kB inhibition, and the suppression of cytokine production by myeloid cells. Changes in the microbial flora can therefore alter not only the local immune environment but also the central nervous system (CNS). SCFAs are thought to have a key role in the bi-directional metabolic pathways between the microbiota, gut, and brain [5]. SCFAs can penetrate the blood–brain barrier using monocarboxylate transporters (MCTs) located on the endothelial cells and then increase the expression of tight junction proteins.

It has now been realized that it is not possible to predict tumor behavior without considering both the immunohistochemical and molecular genetic aspects. While the WHO classification of central nervous system tumors (2016) was the first step in the development of a new approach, the WHO classification (2021) consolidated it [6]. However, the level of the Ki-67 mitotic index remains an important factor in glioma diagnosis [7] as it allows for the differentiation between different degrees of anaplasia in brain tumors. The Ki-67 cell marker, encoded by the *MKI67* gene, is a non-histone protein associated with cell proliferation, and it is actively expressed in the S, G2, late G1, and mitotic phases of cell division. In the morphological diagnosis of central nervous system tumors, the level of this marker is particularly useful for reflecting the proliferative activity of the tumor and distinguishing between low-grade and high-grade gliomas [8,9].

The purpose of this study was to provide a comparative analysis of the major parameters of the blood lipid spectrum in brain tumors and in atherosclerosis as well as to assess any correlation of these indicators with the Ki-67 proliferative activity index of the cells involved.

## 2. Materials and Methods

### 2.1. Materials

Blood plasma was analyzed from 50 patients with brain tumors (25 men aged 32–74 years and 25 women aged 22–81) and from 50 patients with cerebral atherosclerosis (38 men aged 44–79 and 12 women aged 55–77). The cerebral atherosclerosis condition includes chronic elastic and muscular-elastic disease of the arteries of the brain, arising as a result of violations of lipid and protein metabolism, accompanied by the deposition of cholesterol and some fractions of lipoproteins in the intima of the blood vessels with the formation of atheromatous plaques, leading to a slowly progressive disruption of the blood supply to the brain. The brain tumor group had the following tumors predominating: meningiomas (21 patients, 42%), glioblastomas (16 patients, 32%), and astrocytomas (6 patients, 12%); it also included oligodendrogliomas (5 patients, 10%), ependymoma (1 patient, 2%), and gliosarcoma (1 patient, 2%) of varying grades of anaplasia: Grade I—2%, Grade II—8%, Grade III—26%, and Grade IV—64%. Blood plasma from 50 essentially healthy people, in whom brain MRIs revealed no brain tumors, was used to provide comparisons (controls) (16 men aged 24–64 and 34 women aged 24–65). Cerebral atherosclerosis was not detected in either the patient groups with brain tumors or in the healthy individuals (controls). With informed consent, tumor tissues were sampled as postoperative fragments at the Federal State Budgetary Educational Institution of Higher Education, the “Privolzhsky Research Medical University” of the Ministry of Health of the Russian Federation prior to anticancer therapy. Histological diagnoses were made using the WHO classification of CNS tumors [6]. Detailed information regarding the patients is listed in Appendix A. The study was approved by the Ethics Committee of the Federal State Budgetary Educational Institution of Higher Education, the “Privolzhsky Research Medical University” of the Ministry of Health of the Russian Federation (No. 6 of 17.04.2019). Biological objects were obtained during routine treatment procedures with informed consent of patients (Appendix A).

### 2.2. Measurement of Tumor Markers

The postoperative sample material was fixed in 10% formalin for at least 24 h. Basic histological processing of the sampled material was conducted using an Excelsior ES (Thermo Scientific, Waltham, MA, USA), and embedding was performed using a HistoStar station (Thermo Scientific, USA). Each sample with glial neoplasm was immunohistochemically examined to determine the level (high or low) of its Ki-67 proliferative index using SP6 clone (Thermo Scientific, USA). Immunohistochemical staining was performed using the BOND-MAX immunohistostainer and BOND 5.1 software (Leica Biosystems, York, UK), in line with the standard protocols recommended by the manufacturer. Bond polymer refine detection (Leica Biosystems, UK) was the detection system used in the study. The results of the immunohistochemical study were interpreted using light microscopy with a Leica 2500 (Leica Biosystems, UK) equipped with a ×10 magnification eyepiece and ×10, ×20, and ×40 magnification objectives. Cells were considered positively expressed if they had moderate or significant brown staining of their nuclei (Figure 1). The Ki-67 proliferative index was calculated in 10 fields of view at a magnification of ×400; the fields of view was selected using the hot spot method, with a study of at least 1000 cells within the biopsy sample [10]. The proportion of stained tumor cells to all tumor cells in the studied area was calculated. The resulting numerical values were expressed as average percentages across 10 fields of view [11].

### 2.3. Analysis of the Biochemical Parameters of Lipid Metabolism

A determination of the concentration of lipid metabolites was carried out colorimetrically using a UV-mini 1240 spectrophotometer (Shimadzu Corporation, Kyoto, Japan). For the analysis of cholesterol concentrations, a set of cholesterol–UTS reagents (Russia (Cat No. B-11061)) was used. The concentration of triacylglycerols (TAGs) was determined using a set of triglyceride–UTS reagents, Russia (Cat. No. 019.002) for low-density lipoprotein (LDL levels), while high-density lipoprotein (HDL) was determined using a set of HDLC3–HDLC4 reagents (Germany (Cat. No. 17005)) on the StatFaks 04 biochemical analyzer (Awareness Technology, Inc., Palm City, Florida, USA).

### 2.4. Measurement of Levels of Short-Chain Fatty Acids in Blood Samples

Blood from patients was sampled as drops on PerkinElmer 226 blood sample collection filter paper plates, which were then air-dried and stored at −80 °C until analysis.

A Shimadzu LCMS-8050 triple quadrupole liquid chromatograph–mass spectrometer (Shimadzu Corporation, Kyoto, Japan) with a Shimadzu Nexera XR liquid chromatograph system was used for analysis with the LC/MS/MS Method Package for short-chain fatty acids. The package is available online at the following site:

https://www.shimadzu.com/an/products/liquid-chromatograph-mass-spectrometry/lc-ms-system/lcmsms-method-package-for-short-chain-fatty-acids/index.html (accessed on 24 November 2022).

Before analysis, the blood samples were extracted using an internal standard solution (2-ethylbutyric acid in ethanol at 7.92 μmol/L), and 7 calibration solutions of various fatty acid concentrations (from 0.64 nmol/L to 10,000 nmol/L) were prepared and used to construct a calibration curve. Then, the derivatization process was started using a solution of pyridine, ECD (1-ethyl-3-(3-dimethylaminopropyl) carbodiimide hydrochloride), and 3-NPH (3-nitrophenylhydrazine hydrochloride); each resulting mixture was then filtered on a nylon syringe filter with a pore size of 0.22 μm into a vial for the autosampler.

### 2.5. Statistical Analysis

Statistical processing of the data was carried out using StatPlus, version 6, from AnalystSoft Inc. (www.analystsoft.com/ru/). The results of our biochemical research were presented in the form of medians, percentiles, and quartiles (25%; 75%). The significance of the differences obtained was evaluated using non-parametric criteria (Mann–Whitney U-criterion). Correlation analysis between the immunohistochemical marker Ki-67 and the blood lipid metabolism parameters was carried out through the determination of Spearman’s rank coefficient.

## 3. Results

### 3.1. Parameters of Lipid Metabolism Corresponding to Different Pathologies

A (significantly) 63.43% higher level of cholesterol was detected in the blood plasma of patients with brain tumors compared to the level found in patients with cerebral atherosclerosis (Mann–Whitney U-test = 4.78 × 10^−8^) (Table 1).

The levels of LDLs (low-density lipoproteins that are involved in cholesterol transfer into tissues) were also (significantly) 58.70% higher in atherosclerosis compared to their levels in patients with brain tumors (U-criterion = 0.0003) (Table 1).

When comparing the routine parameters of the blood plasma lipidome of the control group and those of patients with atherosclerosis, a statistically significant difference was seen only for the newly considered coefficient, the neutral lipids to cholesterol ratio; thus, the various ratios of these parameters were calculated. Significant differences were found between the values in atherosclerosis and in the “healthy” control group (Table 1, Figure 2).

The analysis of the levels of metabolically dependent compounds, including SCFAs, appears to be a promising approach for studying the blood lipidome in patients with brain tumors (Table 2). Here, it is important to identify the individual person profiles for these compounds in patients with the various types of tumors.

The combination of mass spectrometry and high-performance liquid chromatography allowed for the detection of 12 compounds of interest in the blood samples; these compounds were saturated fatty acids with aliphatic tails containing 1–6 carbon atoms configured as either straight or branched chains (acetate, propionate, butyrate, succinate, isovalerate, etc.). An analysis of the study data showed statistically significant differences in the levels of acetic acid, butyric acid, maleic acid, glyoxylic acid, and pyruvic acid between the “healthy” control group and patients with meningiomas (Table 2). In particular, the analysis of the concentrations of SCFAs in the control group and in the patient group with gliomas demonstrated statistically significant differences in the butyric acid and valeric acid indicators (Table 2). By contrast, a comparison of the meningioma and glioma groups revealed no statistically significant differences between the group values of the tested SCFAs (Table 2).

According to the research data, butyric acid is of the greatest interest among these SCFAs, as significant differences from its blood level in the “healthy” group were found in the meningioma and glioma groups. In particular, the level of butyric acid in both neoplasm groups revealed a sharp decrease compared to the “healthy” control group (Figure 3).

### 3.2. Lipid Metabolism Characteristics Corresponding to the Immunohistochemical Profile of Brain Tumors

To identify the correlations between the Ki-67 marker and the blood lipid spectrum parameters, the Spearman rank correlation coefficient for nonparametric data was calculated (Table 3).

A valid correlation with the Ki-67 level was established for the cholesterol/HDL ratio (Rho = 0.536 and *p* = 0.039) and the neutral lipid/cholesterol ratio (Rho = −0.524 and *p* = 0.039). It is of particular interest that statistically significant correlation coefficients were found for the level of the Ki-67 cell proliferation marker with butyric acid (Rho = −0.526 and *p* = 0.049). Here, the correlation with Ki-67 was seen both in the meningioma group (Rho = −0.516, *p* = 0.05) and in the glioma group (Rho = 0.667, *p* = 0.05).

When the data on the parameters of lipid metabolism were further divided into groups depending on the immunohistochemical profile, taking into account the Ki-67 cell proliferation marker, the figures represented only data with meaningful differences between the groups.

No significant differences in the triacylglycerol/cholesterol ratios of blood plasma were found between the two pathology groups, regardless of whether they had high or low Ki-67 mitotic index levels (Figure 4). However, a significant increase in this ratio was seen in the group with a low Ki-67 level compared to the “healthy” control group (Figure 4).

Reduced butyric acid level in the blood was demonstrated for all levels of the cell proliferation index in the brain tumor tissue when compared to the “healthy” control group. However, no significant differences in the levels of butyric acid in the blood were found between the pathology groups, regardless of whether they had high or low Ki-67 mitotic index levels (Figure 5).

## 4. Discussion

Brain tumors are characterized by their increased cholesterol content, as this supports their growth and proliferation. This increase occurs due to both the activation of its synthesis and the absorption of exogenous cholesterol using lipoprotein receptors for LDLs [12]. The Ki-67 cell proliferative activity indicator is a nuclear protein that participates in ribosomal RNA synthesis [13]. Its expression reflects the degree of cell proliferative activity. In glial neoplasms, Ki-67 is used for differential diagnosis of low-grade (I, II) and high-grade (III, IV) gliomas [14], as its values correlate with the degree of malignancy [15].

Glioblastoma cells accumulate lipid droplets to satisfy their needs for rapid growth. These lipid droplets are subcellular organelles that contain numerous types of neutral lipids, including cholesterol and its esters [16]. As a tumor grows, the permeability of the blood–brain tumor barrier increases [17], resulting in an increase in the level of neutral lipids, particularly cholesterol in the blood.

Thus, the revealed correlation between the ratio of overall neutral lipid/cholesterol in blood plasma and the immunohistochemical marker of cell proliferation activity, the Ki-67 nuclear protein, is very probably due to the accumulation of lipid droplets in the malignant cells and due to the increase in the permeability of the blood–brain barrier resulting from the growth and increasing degree of anaplasia of the brain tumor. In addition to cholesterol levels, our study showed interesting changes in saturated fatty acids, the levels of which were also elevated in the group of patients with brain tumors. Moreover, meningiomas and gliomas showed increases in differing parameters, reflecting the different metabolic activity of these tumor groups due to their different states of histogenesis and grades of malignancy. Cross-changes were found only for butyric acid, which may indicate it as a more versatile metabolite in the energy chain during tumorigenesis. This is also consistent with the correlation of the Ki-67 value with both the cholesterol/HDL and neutral lipid/cholesterol ratios, showing the more active lipid metabolism of tumors that have a high proliferative capacity. Somewhat contradictory results concern other parameters. In particular, the absence of differences in the plasma triacylglycerol/cholesterol ratios between groups for both high and low levels of the Ki-67 mitotic index, and the relatively low LDL values in patients with brain tumors indicate that tumor metabolism is very complex and does not always depend directly on proliferative activity.

It is interesting to consider all the above changes from the point of view of interactions with other body systems, especially with the intestines. SCFAs are major microbial metabolites that primarily serve as an energy supply for colonocytes. Recent studies have confirmed that SCFA levels change in the case of many neurological diseases, including neurodegenerative diseases such as Parkinson’s disease, Alzheimer’s disease, cerebrovascular diseases (stroke, transient ischemic attack), epilepsy, and neuroimmune inflammatory diseases (multiple sclerosis), autism spectrum disorder, and depressive disorders, hence implying that SCFAs may be vitally important for the microbiota–gut–brain axis connections [18].

The gut–brain axis is a newly discovered and under-investigated phenomenon, the understanding of which might be used to improve the efficacy of anti-glioma therapy [19]. Among SCFAs, the most common ones are acetate, propionate, and butyrate [20], together accounting for 95% of all SCFAs [21].

SCFAs penetrate into endothelial cells using MCTs. In human and rodent brain endothelium, MCT1, acting as a proton-dependent cotransporter/exchanger, has a key role in the entry of SCFAs into the brain parenchyma [22]. By forming a special pool of metabolites, malignant neoplasms can probably modulate the production of neurotransmitters using intestinal nerve terminals and their de novo synthesis. Dono A, Patrizz A, McCormack RM et al. (2020) [23] demonstrated that the levels of various SCFAs and neurotransmitters in gliomas change in both experiments in mice and in human patients. Changes in the pool of SCFAs after glioma development seem to be related to CNS damage and to blood–brain barrier destruction, which result in changes in the microbial community through the gut axis interaction, leading to further reduction in bacterial SCFA production [23].

Blood–brain barrier destruction mechanisms may include the loss of tight junction integrity, increased transcytosis, endothelial cell apoptosis, the disruption of the glial limiting membrane, glycocalyx degradation, etc.

Some researchers have suggested that SCFAs function in the brain by means of histone deacetylase (HDAC) inhibition, with butyrate being the most potent class I and class IIa HDAC inhibitor [24]. In the central nervous system, such SCFA inhibition of HDAC histone deacetylase activity increases the acetylation of lysine residues in the nucleosomal histones. These changes in acetylation regulate the genesis and development of carcinomas [25].

Several publications have demonstrated evidence that SCFAs are associated with intracellular cascades that have the effect of helping to repair the impaired blood–brain barrier by increasing the expression of proteins of the intercellular junctional complex [26,27]. According to other studies, SCFA-induced restoration of the integrity of the blood–brain barrier can be mediated by the direct impact of SCFAs on brain endothelial cells, as well as by the suppression of systemic inflammation and the inhibition of activation of the microglia and astrocytes [28,29].

Microglia are very sensitive to SCFAs. Taking into account that three SCFA-specific receptors—GPR43, GPR41, and GPR109A—are expressed in microglial cells and that HDAC involvement in microglial activation has been demonstrated, the SCFA–microglia interaction seems to be even more important for blood–brain barrier preservation than the direct action of SCFAs on endothelial cells [30].

Butyrate also inhibits STAT3 signaling, thus suppressing the expression of bcl-2, bcl-XL, c-myc, cyclin D1, and HIF-1, resulting in decreased cell proliferation and increased apoptosis in both hypoxia and in colon cancer [31]. SCFAs also enhance the methylation of oncogenes, thereby decreasing their expression [32].

Ras signaling activation of STAT3 and the ability of SCFAs strongly to inhibit Ras activity may also reduce STAT3 activation, resulting in decreased expression of the proinflammatory IL-8 and VEGF interleukins and the blocking of angiogenesis [33]. For instance, strong anti-inflammatory effects of SCFAs have been shown particularly for butyrate through the inhibition of the IFNγ signaling pathway and the blocking of JAK2/STAT1 [34,35].

Thus, SCFAs are integral parts of the metabolic cascades of both normal and neoplastic tissues; they form a pool of regulatory molecules involved in neuroendocrine, immune, and neural pathways, requiring more detailed study for prognosis and treatment purposes.

It is important to note several possible limitations of our study. We understand that the lipid spectrum can be influenced by many factors, and we had therefore tried to take these into account as much as possible when selecting patients. However, individual characteristics, such as genetic predisposition, lifestyle, bad habits, and diet variability present difficulties in this respect. Therefore, it is necessary to consider that some values may fluctuate slightly due to the influence of these factors. It is also necessary to realize the potential impact of the heterogeneity of tumors, not only relative to each other, but even within a single tumor. For example, some gliomas can have areas with different levels of proliferation, which can therefore cause difficulties in interpreting the Ki-67 values. We aimed to include the maximum possible number of cases available to our study in order to minimize the influence of random factors, but a larger study will be required for determining patterns with greater certainty. In the future, we plan to expand our study to include a larger number of patients and to incorporate data that could indicate the degree of influence of other relevant factors.

## 5. Conclusions

The study identified several parameters of the lipid spectrum in blood plasma, the levels of which significantly change as a result of brain tumors and differ from the lipidome changes seen with atherosclerosis. The ratio of neutral lipids to cholesterol can be determined on the basis of routine clinical tests and thus be used for preoperative diagnosis of brain tumors. A decreased level of butyric acid in the blood can be an additional verifying criterion for brain tumor diagnosis, as it is especially significant in the initial stages of anaplasia. However, these parameters cannot serve as a preoperative non-invasive predictor of the level of the Ki-67 mitotic index, as no significant differences between them were found for low-grade and high-grade anaplasia of brain tumors.

## Figures and Tables

**Figure 1 cimb-47-00228-f001:**
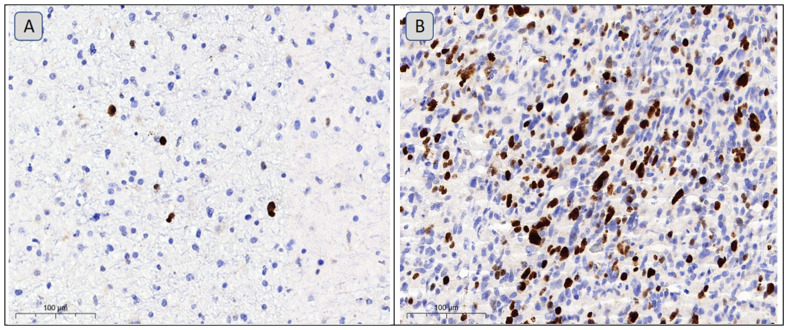
Immunohistochemical study of the Ki-67 marker. Magnification ×200; Bar: 100 µm. The images were used to determine the level of nuclear non-histone protein; the Ki-67 marker of cell proliferation in the material were obtained from gliomas. (**A**) The locally pronounced but overall limited brown staining of nuclei (up to 5% of cells in a few hot spots) shows the low Ki-67 proliferation index in a typical Grade II oligodendroglioma. (**B**) The diffuse but pronounced nuclear brown staining (up to 50% of cells in hot spots) shows a high Ki-67 proliferation index in a Grade IV glioblastoma.

**Figure 2 cimb-47-00228-f002:**
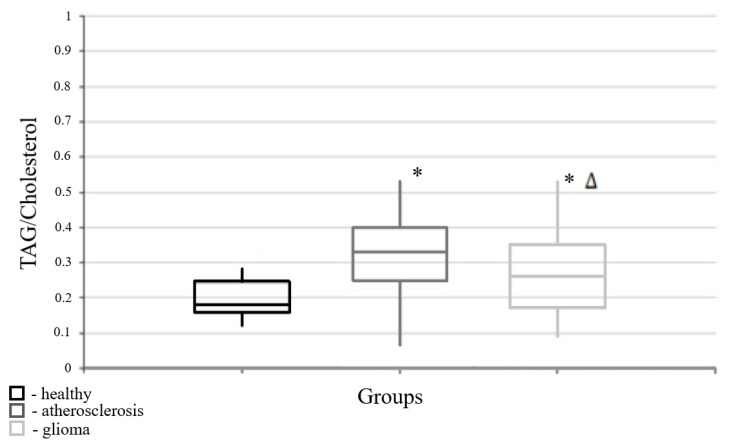
Medians and interquartile ranges (IQRs) of the triacylglycerol/cholesterol ratios in the blood plasma of patients with either atherosclerosis or brain tumors. There are significant differences in the values of the triacylglycerol/cholesterol ratios of blood plasma between the atherosclerosis and brain tumor groups (U-criterion of Mann–Whitney *p* = 0.046) and the brain tumors/healthy persons (U-criterion of Mann–Whitney *p* = 0.016). Legend: *—statistically significant differences from the “healthy” group (*p* < 0.05); Δ—statistically significant differences from the atherosclerosis group (*p* < 0.05).

**Figure 3 cimb-47-00228-f003:**
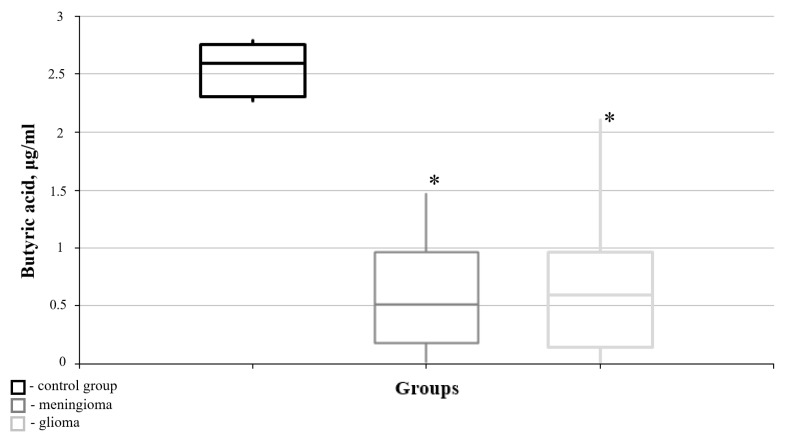
Medians and interquartile ranges (IQRs) of blood butyric acid concentration in the meningioma and glioma groups. There are significant differences in the blood butyric acid content between the meningioma and “healthy” groups (U-criterion of Mann–Whitney *p* = 0.016) and between the glioma and “healthy” groups (U-criterion of Mann–Whitney *p* = 0.004). Legend: *—statistically significant differences from the “healthy” group (*p* < 0.05).

**Figure 4 cimb-47-00228-f004:**
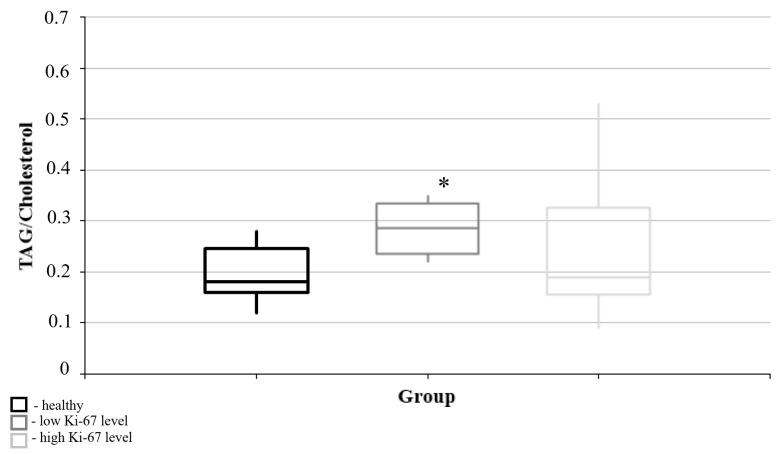
Medians and interquartile ranges (IQRs) of the triacylglycerol/cholesterol ratio in the blood plasma in relation to the magnitude of the Ki-67 mitotic index. There is a significant difference in the triacylglycerol/cholesterol ratio between groups with both low and high Ki-67 mitotic indices (U-criterion of Mann–Whitney *p* = 0.05). Legend: *—statistically significant differences from the “healthy” group (*p* < 0.05).

**Figure 5 cimb-47-00228-f005:**
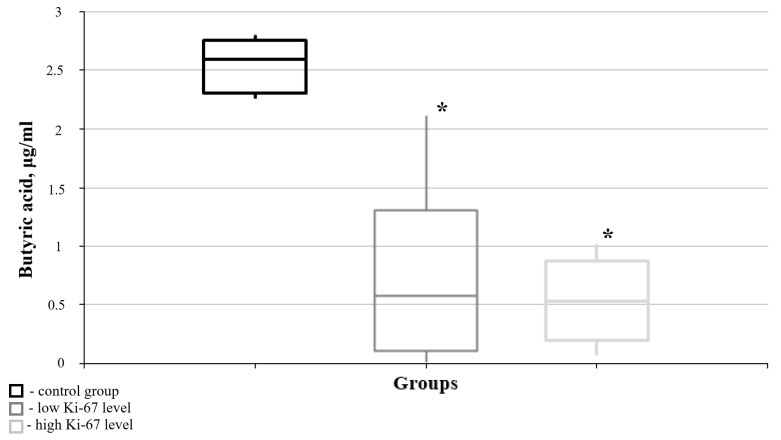
Medians and interquartile ranges (IQRs) of the content of butyric acid in the blood in the case of brain tumors as functions of the value of the Ki-67 mitotic index. There are significant differences between the butyric acid levels in the blood of the “healthy” controls and the low level of marker tumors (Mann–Whitney U-test: *p* = 0.008) and for the high Ki-67 mitotic index groups (Mann–Whitney U-test: *p* = 0.016). Legend: *—statistically significant differences from the “healthy” group (*p* < 0.05).

**Table 1 cimb-47-00228-t001:** Indicators of lipid metabolism found in blood plasma from patients with brain tumors or atherosclerosis.

Parameters	Healthy Persons		Brain Tumors		Atherosclerosis		U-Criterion of Mann–Whitney Test		
	Median	Quartiles	Median	Quartiles	Median	Quartiles	Difference Between Healthy Persons and Brain Tumors	Difference Between Healthy Persons and Atherosclerosis	Difference Between Brain Tumors and Atherosclerosis
Cholesterol	4.6	(4.5; 4.7)	5.72	(4.88; 6.415)	3.5	(2.9; 4.4)	0.316	0.002 *	4.78 × 10^−8^ *
Triacylglycerols	0.8	(0.765; 2.01)	1.43	(1.01; 2.02)	1.21	(0.87; 1.59)	0.138	0.714	0.248
Low-density lipoproteins	2.9	(2.8; 2.85)	3.65	(2.67; 4.30)	2.30	(2.1; 2.7)	0.225	0.001 *	0.0003 *
High-density lipoproteins	1.56	(1.43; 1.755)	1.34	(1.22; 1.61)	1.15	(0.945; 1.42)	0.497	0.139	0.135
Triacylglycerols/cholesterol	0.18	(0.16; 0.255)	0.28	(0.17; 0.35)	0.33	(0.25; 0.4)	0.016 *	0.0001 *	0.046 *
Triacylglycerols/Low-density lipoproteins	0.32	(0.27; 0.42)	0.4 (0.31; 0.56)	(0.495; 0.285)	0.03	(0.395; 0.69)	0.089	0.002 *	0.039 *
Cholesterol/High-density lipoproteins	2.965	(2.73; 3.395)	4.34 (2.86; 5.76)	(4.08; 2.735)	2.72	(2.3; 3.33)	0.129	0.158	0.007 *
Low-density lipoproteins/High-density lipoproteins	0.705	(0.56; 1.14)	2.88 (1.79; 3.61)	(4.08; 2.735)	1.52	(1.035; 2.36)	0.003 *	0.0005 *	0.108
Triacylglycerols/High-density lipoproteins	0.55	(0.46; 0.89)	1.54	(1.54; 1.065)	0.82	(0.7; 1.29)	0.011 *	0.019 *	0.769
Cholesterol/Low-density lipoproteins	1.72	(1.61; 1.855)	1.315	(1.315; 0.725)	1.865	(1.49; 2.17)	0.085	0.671	0.213

Legend: *—statistically significant differences.

**Table 2 cimb-47-00228-t002:** Concentrations of short-chain fatty acids in the blood of patients with brain tumors.

	Healthy Persons (0)		Meningiomas (1)		U-Criterion of Mann–Whitney	Gliomas (2)		U-criterion of Mann–Whitney (Difference from the “Healthy” Control Group)	U-criterion of Mann–Whitney (Difference from the “Healthy” Control Group)
Acid	Median	Quartiles	Median	Quartiles		Median	Quartiles		
		Q1		Q1			Q1		
		Q2		Q2			Q2		
		Q3		Q3			Q3		
Lactic acid	268.1	237.7	554.4	298.5	0.142	439.1	313.4	0.028	1.000
		268.1		554.4			439.1		
		303.3		640.8			779.1		
Acetic acid	68.28	64.95	30.78	11.08	0.014 *	39.43	31.64	0.361	0.394
		68.28		30.78			39.43		
		76.00		55.50			99.93		
Propionic acid	3.880	2.160	3.177	1.489	0.624	3.247	1.983	0.273	1.000
		3.880		3.177			3.247		
		5.614		3.956			3.906		
Isobutyric acid	0.727	0.669	0.610	0.349	0.806	0.915	0.254	1.000	0.670
		0.727		0.610			0.915		
		0.816		0.990			3.230		
Butyric acid	2.592	2.289	0.515	0.075	0.014 *	0.591	0.064	0.006 *	0.748
		2.592		0.515			0.591		
		2.778		1.303			1.283		
Succinic acid	0.240	0.199	0.477	0.243	0.221	0.377	0.192	0.361	0.670
		0.240		0.477			0.377		
		0.295		0.839			0.646		
Isovaleric acid	0.192	0.116	0.211	0.019	1.000	0.272	0.199	0.272	0.915
		0.192		0.211			0.272		
		0.266		0.581			0.325		
Valeric acid	2.700	2.633	0.670	0.014	0.014	1.169	0.587	0.006 *	0.394
		2.700		0.670			1.169		
		2.753		1.292			1.854		
Maleic acid	49.26	32.43	5.730	3.737	0.014 *	1.906	0.000	0.360	0.521
		49.26		5.730			1.906		
		61.98		16.01			143.3		
Glyoxylic acid	0.390	0.379	0.944	0.738	0.014 *	1.557	0.491	0.067	0.670
		0.390		0.944			1.557		
		0.397		1.745			2.211		
Pyruvic acid	5.350	3.916	2.722	1.965	0.027 *	4.012	3.092	0.584	0.088
		5.350		2.722			4.012		
		6.463		4.001			7.373		
Glycolic acid	1.275	0.808	2.848	1.419	0.086	1.524	0.954	0.361	0.286
		1.275		2.848			1.524		
		1.652		6.029			2.782		

Legend: *—statistically significant differences from the “healthy” group.

**Table 3 cimb-47-00228-t003:** Correlations (Spearman’s rank coefficient) between the immunohistochemical Ki-67 marker of cell proliferation and lipid metabolism parameters of blood.

Parameter of Lipid Metabolism of Blood Plasma	Statistical Parameters (Spearman’s Coefficient)	Ki-67 Marker of Cell Proliferation in Glioblastoma Material
Cholesterol	Rho	0.290
	p	0.191
Triacylglycerols	Rho	−0.169
	p	0.478
Low-density lipoproteins	Rho	0.155
	p	0.567
High-density lipoproteins	Rho	−0.222
	p	0.409
Triacylglycerols/Cholesterol	Rho	−0.524 *
	p	0.039
Triacylglycerols/Low-density lipoproteins	Rho	−0.412
	p	0.127
Cholesterol/High-density lipoproteins	Rho	0.536 *
	p	0.039
Low-density lipoproteins/High-density lipoproteins	Rho	0.381
	p	0.179
Triacylglycerols/High-density lipoproteins	Rho	0.172
	p	0.494
Cholesterol/Low-density lipoproteins	Rho	−0.014
	p	0.956
Lactic acid	Rho	0.455
	p	0.137
Acetic acid	Rho	−0.158
	p	0.6247
Propionic acid	Rho	−0.322
	p	0.307
Isobutyric acid	Rho	0.112
	p	0.729
Butyric acid	Rho	−0.526 *
	p	0.044
Succinic acid	Rho	0.388
	p	0.212
Isovaleric acid	Rho	−0.123
	p	0.703
Valeric acid	Rho	−0.463
	p	0.130
Maleic acid	Rho	−0.358
	p	0.253
Glyoxylic acid	Rho	0.431
	p	0.162
Pyruvic acid	Rho	0.315
	p	0.318
Glycolic acid	Rho	0.237
	p	0.458
Malic acid	Rho	−0.541
	p	0.069

Legend: *—statistically significant differences from the “healthy” group (*p* < 0.05).

## Data Availability

The original contributions presented in this study are included in the article/Appendix A. Further inquiries can be directed to the corresponding author(s).

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
