# Peer review of "Brain Tumor-Induced Changes in Routine Parameters of the Lipid Spectrum of Blood Plasma and Its Short-Chain Fatty Acids"

_cimb, 2025, doi:10.3390/cimb47040228_

Round 1
Reviewer 1 Report
Comments and Suggestions for Authors
The study is designed to measure various parameters of lipid spectrum in blood plasma and they see significant changes in patients with brain tumor. The study is executed well but I have major concern about how lifestyle, diet, sex and age can affect lipid spectrum in blood plasma?
Author Response
Comments 1: [The study is designed to measure various parameters of lipid spectrum in blood plasma and they see significant changes in patients with brain tumor. The study is executed well but I have major concern about how lifestyle, diet, sex and age can affect lipid spectrum in blood plasma?]
Response 1: Thank you very much. Yes, these parameters may influence the lipid profile in blood plasma to some extent. In our study, we did not include an assessment of the degree of influence of these factors, but we did aim to select patients without any obvious lifestyle or dietary features that could significantly affect the lipid profile. Gender and age may create some deviations, but these should not be overly significant when interpreting the results. In future studies, we will try to expand our sample and take these parameters into account as well.
Reviewer 2 Report
Comments and Suggestions for Authors
Comments
Materials
In line 67, please define the term “cerebral atherosclerosis”.
Please clarify whether patients with brain tumors, and healthy people had cerebral atherosclerosis or not.
In line 78, you stated “Histological diagnoses were made using the WHO classification of CNS tumors [6]”. However, reference [7] would be more appropriate.
In line 108, “The concentration of neutral lipids (TAG)” should be corrected because TAG does not stand for concentration of neutral lipids.
Statistical analysis
In line 131-132, clarify which results are presented in the form of medians, percentiles and quartiles (25%; 75%).
Which data were used for Mann-Whitney U test and correlation analysis?
Results
In line 143, “in atherosclerosis” should be replaced with another term, such as “in patients with cerebral atherosclerosis”, which has appeared in line 67.
In Table1, provide abbreviation of each word, such as LDL, HDL, and TAG.
In line 178, “abrupt” should be replaced with “sharp”, which makes the sentence smoother.
Discussion
This section focused too much on the explanation of the basic knowledge rather than interpretation of the results.
Regarding line 232-236, as a tumor grows, how do lipid droplets contribute to the increase in the level of neutral lipids and cholesterol in the blood?
In line 239, “very” should be omitted.
In line 37, 242, 253, 259 and 294, spelling out “short-chain fatty acids” is unnecessary because it has already appeared in line 36.
In line 253-254, spelling out “monocarboxylate trans porters” is unnecessary because it has already appeared in line 47-48.
Comments on the Quality of English LanguageSome words or terms that are not appropriate for the context were found as follows.
In line 108, “The concentration of neutral lipids (TAG)” should be corrected because TAG does not stand for concentration of neutral lipids.
In line 143, “in atherosclerosis” should be replaced with another term, such as “in patients with cerebral atherosclerosis”, which has appeared in line 67.
In line 178, “abrupt” should be replaced with “sharp”, which makes the sentence smoother.
In line 239, “very” should be omitted.
Author Response
Materials
Comments 1: [In line 67, please define the term “cerebral atherosclerosis”.]
Response 1: Thank you very much for your comment. (Added to text.)
Comment 2: [Please clarify whether patients with brain tumors, and healthy people had cerebral atherosclerosis or not.]
Response 2: Cerebral atherosclerosis was not detected in patients with brain tumors and healthy individuals in the study groups. (Added to text).
Comments 3: [In line 78, you stated “Histological diagnoses were made using the WHO classification of CNS tumors [6]”. However, reference [7] would be more appropriate.]
Response 3: We understand from the context that [6] should be moved to [7], and not vice versa.
Comments 4: [In line 108, “The concentration of neutral lipids (TAG)” should be corrected because TAG does not stand for concentration of neutral lipids.]
Response 4: Thank you very much for your comment. Added to text.
Statistical analysis
Comments 5: [In line 131-132, clarify which results are presented in the form of medians, percentiles and quartiles (25%; 75%).]
Response 5: Thank you very much for your comment. Added to text.
Comments 6: [Which data were used for Mann-Whitney U test and correlation analysis?]
Response 6: Thank you very much for your comment. Added to text.
Results
Comments 7: [In line 143, “in atherosclerosis” should be replaced with another term, such as “in patients with cerebral atherosclerosis”, which has appeared in line 67.]
Response 7: There is probably a mistake here, because line 143 doesn't say anything about atherosclerosis at all. Apparently line 151 was meant. Replaced. (Added to text)
Comments 8: [In Table1, provide abbreviation of each word, such as LDL, HDL, and TAG.]
Response 8: Thank you very much for your comment. (Added to text.)
Comments 9: [In line 178, “abrupt” should be replaced with “sharp”, which makes the sentence smoother.]
Response 9: Thank you very much for your comment. (Added to text), but not on line 178, but on line 196
Discussion
Comments 10: [This section focused too much on the explanation of the basic knowledge rather than interpretation of the results.]
Response 10: Added one paragraph to the discussion with a more detailed discussion of the results. (Added to text.)
Comments 11: [Regarding line 232-236, as a tumor grows, how do lipid droplets contribute to the increase in the level of neutral lipids and cholesterol in the blood?]
Response 11: Thank you very much for your comment. We have taken it into account. (Added to text.)
Comments 12: [In line 239, “very” should be omitted.]
Response 12: There is no word very in line 239. Apparently they were talking about line 257. Removed.
Comments 13: [In line 37, 242, 253, 259 and 294, spelling out “short-chain fatty acids” is unnecessary because it has already appeared in line 36.]
Response 13: Other lines are indicated. Replaced everywhere with the abbreviation after the first use.
Comments 14: [In line 253-254, spelling out “monocarboxylate trans porters” is unnecessary because it has already appeared in line 47-48.]
Response 14: Thank you very much for your comment. We have taken it into account. (Added to text.)
Comments 15: [In line 108, “The concentration of neutral lipids (TAG)” should be corrected because TAG does not stand for concentration of neutral lipids.]
Response 15: Thank you very much for your comment. We have taken it into account. (Added to text.)
Comments 16: [In line 143, “in atherosclerosis” should be replaced with another term, such as “in patients with cerebral atherosclerosis”, which has appeared in line 67.]
Response 16: Thank you very much for your comment. We have taken it into account. (Added to text.)
Comments 17: [In line 178, “abrupt” should be replaced with “sharp”, which makes the sentence smoother.]
Response 17: Thank you very much for your comment. We have taken it into account. (Added to text.)
Comments 18: [In line 239, “very” should be omitted.]
Response 18: There is no word very in line 239. Apparently they were talking about line 257. Removed.
Reviewer 3 Report
Comments and Suggestions for Authors
The article is written in a competent manner. It introduces the reader to the subject in a satisfactory manner, contains a properly formulated research hypothesis, and describes the methods employed in a satisfactory manner. The results section is described in a sufficiently thorough manner.
Although the discussion is generally well written, it could be improved by a more precise and organised presentation of the data. It would be beneficial to make a clear distinction between the studies that support the findings and those that contradict them. This will help readers to understand the broader context of the research and its place within the existing body of knowledge. In addition, the inclusion of a section on the limitations of the study would provide a more balanced and critical perspective, acknowledging any limitations or potential weaknesses in the research and suggesting directions for future work.
The article under consideration is well-structured and contributes significantly to the field. It explores an important and relatively under-researched area, focusing on the relationship between lipid metabolism, short-chain fatty acids, and brain tumours, particularly their correlation with the Ki-67 proliferation index. The study provides valuable insights into potential preoperative diagnostic markers and metabolic changes associated with brain tumours. In the context of future studies, it would be worthwhile to expand the research group in order to confirm the obtained results in a larger cohort of patients. Moreover, the study could benefit from a longitudinal design in order to observe changes in lipid markers over time, particularly before and after treatment. The conclusions drawn from the conducted studies are accurate and consistent with the obtained results and supported by the latest scientific literature.
Author Response
Comments 1: [The article is written in a competent manner. It introduces the reader to the subject in a satisfactory manner, contains a properly formulated research hypothesis, and describes the methods employed in a satisfactory manner. The results section is described in a sufficiently thorough manner.
Although the discussion is generally well written, it could be improved by a more precise and organised presentation of the data. It would be beneficial to make a clear distinction between the studies that support the findings and those that contradict them. This will help readers to understand the broader context of the research and its place within the existing body of knowledge. In addition, the inclusion of a section on the limitations of the study would provide a more balanced and critical perspective, acknowledging any limitations or potential weaknesses in the research and suggesting directions for future work.]
Response 1: We added one paragraph to the discussion with reasoning about the results. And at the end we added a paragraph about possible limitations.
Comments 2: [The article under consideration is well-structured and contributes significantly to the field. It explores an important and relatively under-researched area, focusing on the relationship between lipid metabolism, short-chain fatty acids, and brain tumours, particularly their correlation with the Ki-67 proliferation index. The study provides valuable insights into potential preoperative diagnostic markers and metabolic changes associated with brain tumours. In the context of future studies, it would be worthwhile to expand the research group in order to confirm the obtained results in a larger cohort of patients. Moreover, the study could benefit from a longitudinal design in order to observe changes in lipid markers over time, particularly before and after treatment. The conclusions drawn from the conducted studies are accurate and consistent with the obtained results and supported by the latest scientific literature.]
Response 2: Thank you very much. Yes, in the future we plan to expand the sample and take into account more factors, including assessing dynamic changes.
Round 2
Reviewer 1 Report
Comments and Suggestions for Authors
The manuscript can be accepted.
Reviewer 2 Report
Comments and Suggestions for Authors
The authors have addressed all my concerns about this article. I have no further comments.